# Alkaloid Profiling, Anti-Enzymatic and Antiproliferative Activity of the Endemic Chilean Amaryllidaceae *Phycella cyrtanthoides*

**DOI:** 10.3390/metabo12020188

**Published:** 2022-02-18

**Authors:** Carlos Fernández-Galleguillos, Javier Romero-Parra, Adrián Puerta, José M. Padrón, Mario J. Simirgiotis

**Affiliations:** 1Instituto de Farmacia, Facultad de Ciencias, Universidad Austral de Chile, Campus Isla Teja, Valdivia 5090000, Chile; 2Departamento de Química Orgánica y Fisicoquímica, Facultad de Ciencias Químicas y Farmacéuticas, Universidad de Chile, Olivos 1007, Casilla 233, Santiago 6640022, Chile; javier.romero@ciq.uchile.cl; 3BioLab, Instituto Universitario de Bio-Orgánica Antonio González (IUBO-AG), Universidad de La Laguna, 38206 La Laguna, Spain; apuertaa@ull.es (A.P.); jmpadron@ull.es (J.M.P.); 4Center for Interdisciplinary Studies on the Nervous System (CISNe), Universidad Austral de Chile, Valdivia 5090000, Chile

**Keywords:** Amaryllidaceae alkaloids, *Phycella*, cholinesterase, tyrosinase, antiproliferative, UHPLC-MS

## Abstract

This research aims to identify the alkaloid profile and to evaluate the enzyme inhibitory potential and antiproliferative effects of the Amaryllidaceae plant *Phycella cyrtanthoides*. The alkaloid extracts from bulbs and leaves were analyzed using ultrahigh performance liquid chromatography orbitrap mass spectrometry (UHPLC-Orbitrap-MS) analysis. A total of 70 alkaloids were detected in the *P. cyrtanthoides’* extracts. The enzyme inhibition potential against cholinesterases (AChE: acetylcholinesterase, and BChE butyrylcholinesterase) and tyrosinase were studied. Bulbs displayed the best IC_50_ values against AChE (4.29 ± 0.03 µg/mL) and BChE (18.32 ± 0.03 µg/mL). These results were consistent with docking experiments with selected major compounds in the active sites of enzymes, while no activity was observed against tyrosinase enzyme. Antiproliferative effects were investigated against human cervical (HeLa), lung (A549, SW1573), colon (WiDr), and breast (HBL-100, T-47D) tumor cell lines. Bulbs and leaves were active in all cell lines (GI_50_ < 2.5 µg/mL). These findings suggest that the endemic Chilean plant *P. cyrtanthoides* contains diverse types of bioactive alkaloids with antiproliferative activities and inhibitory effects with potential therapeutic applications for neurodegenerative diseases

## 1. Introduction

Plants belonging to the Amaryllidaceae family are known for the biosynthesis of pharmacologically active alkaloids [1]. Traditionally, plants extracts of this family have been used as folk medicine for cancer in Ancient Greece, Asia, Africa, and Polynesia for a variety of ailments [2,3]. Galanthamine, an acetylcholinesterase (AChE) inhibitor, is well known for being the most important Amaryllidaceae alkaloid (AA) extracted and the first commercial natural product for the treatment of Alzheimer’s disease (AD). The inhibition of AChE enzyme restored the levels of acetylcholine (ACh) in the postsynaptic neuronal membrane, improving the decline of cognitive function. Similarly, butyrylcholinesterase (BChE) enzyme also has an important function in cholinergic transmission and their levels are increased in AD [4]. Thus, several research groups have focused on finding new sources of bioactive alkaloids from Amaryllidaceae plants with cholinesterase inhibitory potential. On the other hand, these plants showed strong antiproliferative activity. Amaryllidaceae alkaloids, such as lycorine, haemantamine, pancratistatine, and montanine, have been extensively screened for their antiproliferative effects [3,5,6,7,8]. Considering the structural variety and pharmacological properties of AAs, further studies aiming to identify and characterize active compounds would contribute to optimize their therapeutic applications. Different analytical methods have been used for the analysis of AAs, including thin layer chromatography (TLC) [9], capillary electrophoresis (CE) [10], and capillary-electrophoresis-MS (CE-MS) [11]. However, GC-MS and HPLC-MS have been widely and successfully employed in the analysis of AAs from plant sources [12,13,14,15]. Recently, ultra-high-performance liquid chromatography quadrupole time-of-flight mass spectrometry (UHPLC-QTOF-MS) has been used for the analysis of crude extracts in different *Lycoris* species [16].

The Amaryllidaceae family comprises 85 genera and approximately 1100 species, which are widely distributed in tropical, subtropical, and warm regions around the world [17]. In Chile, approximately 9 genera and 45 species have been described [18]. Additionally, some species belonging to the *Traubia*, *Placea* and *Phycella* genera have been cataloged as endemic [19,20].

*Phycella* genus is distributed in central Chile (from the Coquimbo to Bíobío region), where the following five species have been identified: *P. australis*, *P. scarlatina*, *P. herbetiana* (also present in Argentina), *P. brevituba,* and *P. cyrtanthoides* [21]. *P. cyrtanthoides* (local name: Añañuca de Fuego) is an endemic plant mainly found in the Metropolitana (Santiago) region and characterized by having a paraperigonium with fimbriae and six red flowers for umbel (Figure 1). To the best of our knowledge, no reports have been published that describe the chemistry and pharmacological properties of *P. cyrtanthoides* plants.

In previous reports, the bulbs of *P. australis* were analyzed by GC-MS and showed a high content of haemantamine-type alkaloids. Pharmacological studies of the alkaloid fractions on human neuroblastoma cells suggested remarkable neuroprotective properties [22]. This study is one of the few regarding the chemical and pharmacological properties of the Chilean *Phycella*. The alkaloid profile and the AChE inhibitory potential from bulbs of the Argentinian *P. herbetiana* have also been investigated [23].

Considering the potential of AAs, we decided to explore the chemistry and pharmacological properties of *P. cyrtanthoides* for the first time using UHPLC-Orbitrap-MS. We screened the biological properties of the extracts based on enzymatic and cell models. The main objective of the present study is to investigate the alkaloid profile from the endemic *Phycella cyrtanthoides* (Amaryllidaceae) from bulbs and leaves organs. The alkaloid-rich extracts were subjected to cholinesterase and tyrosinase inhibitory evaluation. Selected major compounds were studied by molecular docking to investigate intermolecular interactions with cholinesterase and tyrosinase enzymes. Antiproliferative activities were also evaluated using a panel of six solid tumor cells lines.

## 2. Results and Discussion

### 2.1. Alkaloid Profiling of Phycella Cyrtanthoides Extracts

In the present work, the alkaloid profiling of bulbs and leaves from the methanolic extracts of *Phycella cyrtanthoides* was investigated. This is the first report concerning the alkaloid profiling of *P. cyrtanthoides*. In total, 70 alkaloids were detected by UHPLC-PDA-Orbitrap-Mass Spectrometry, among them 47 were tentatively identified and 23 could not be identified using the techniques described herein (see Table 1, and Figure 2). One alkaloid was identified by spiking experiments with available standards (lycorine hydrochloride) using positive mode of detection. The generation of molecular formulas was performed using high resolution accurate mass analysis (HRAM) and matching with the isotopic pattern. Finally, analyses were confirmed using MS/MS data, fragmentation pattern, database of MassBank of North America (MoNA), and the published literature.

Lycorin and homolycorine-type alkaloids were the most common alkaloids identified on the extracts. In addition, crinine, haemanthamine, tazettine and belladine-type alkaloids were also detected. The distribution of the alkaloids in *P. cyrtanthoides* varied according to the organs. However, a total of nine alkaloids (lycorine, pseudolycorine, hippeastrine, candimine, haemanthamine, vittatine, 7-methoxy-*O*-methyllycorenine, 5-methyl-2-epihippamine and 1-*O*-acetylbuphanamine) were found both in bulbs and leaves (Figure 3). The representative interpretations among the identified alkaloids are discussed below.

Peak 1 with a [M+H]^+^ ion at *m/z* = 362.13348 was identified as 2-hydroxyalbomaculine, previously isolated from the aerial parts of *Zephyranthes candida* [24]. Peak 2 was identified as tazettine (C_18_H_22_O_5_N^+^, 332.15756), which was reported previously from bulbs of Chilean *Phycella australis* [22] and the Argentinian *Phycella herbetiana* [23]. Peak 3 was identified as powelline (C_17_H_20_O_4_N^+^, 302.14523) based on the formation of [M+H-H_2_O]^+^ ion at *m/z* = 284.13364, suggesting the presence of hydroxyl group at C-3 [25]. Peak 4 with a [M+H]^+^ ion at *m/z* = 288.12869 and diagnostic fragments formed after the retro Diels–Alder rearrangements (RDAr) at *m/z* = 239.05540, *m/z* = 147.04440, and *m/z* = 119.04981 was identified as lycorine (C_16_H_18_O_4_N^+^) [23]; peak 5 with a *m/z* = 290.14432 was identified as pseudolycorine (C_16_H_20_O_4_N^+^) according to the fragmentation data [26] and previously reported from *P. australis* [22] *and P. herbetiana* [23].

Peak 6 with a [M+H]^+^ ion at *m/z* = 316.12579 showed diagnostic fragments detected at *m/z* = 298.11389 (loss of H_2_O), *m/z* = 280.10153 (loss of 2H_2_O) and *m/z* = 191.03445 (formed by RDAr) in agreements with hippeastrine (C_17_H_18_O_5_N^+^). Peak 7 and peak 8 produced the same fragmentation ions and were identified as hippeastrine isomers [27]. Peak 9 was identified as pluviine (C_17_H_22_O_3_N^+^, 288.12869), peak 10 as dihydrolycorine (C_16_H_20_O_4_N^+,^ 290.14432) and also detected on *Phycella australis* [22], and peak 11 as 3-*O*-methyl-epimacowine (C_16_H_18_O_4_N^+^, 288.12872), previously identified from the bulbs of the Brazilian *Hippeastrum calyptratum* [28]. Peak 12 with a [M+H]^+^ ion at *m/z* = 316.12589 was identified as 11-*O*-methylcrinamine [29], peak 13 as 3-hydroxydihydrocaranine (C_16_H_20_O_4_N^+^, 290.14420), and peaks 14 and 15 were proposed as epi-zephyranthine isomers (C_16_H_20_O_4_N^+^, 290.14420). Peak 16 and peak 17 were identified as hippeastrine isomers, while peak 18 with a [M+H]^+^ ion at *m/z* = 346.13831 was identified as candimine, previously reported from *Hippeastrum morelianum* bulbs [30]. Peak 19 with a [M+H]^+^ ion at *m/z* = 274.14804 was identified as 8-*O*-demethylmaritidine [31] and peak 20 with fragments at *m/z* = 272.13251, *m/z* = 180.10258 and *m/z* = 167.11855 was identified as haemanthamine (C_17_H_20_O_4_N^+^, 302.14548). Haemanthamine was also identified on *Phycella australis* [22].

Peak 21 was identified as 5-methyl-epimethylpseudolycorine [16], peak 22 as 2α-methoxy-6-*O*-ethyloduline previously isolated from *Lycoris radiata* [32], peak 23 with diagnostic fragments at *m/z* = 180.10249 and *m/z* = 167.01340 was proposed as vittatine (C_16_H_18_O_3_N^+^, 272.13257) previously detected from *P. australis* [22] and *P. herbetiana* [23], peak 24 as 10-norpluviine (C_16_H_20_O_3_N^+^, 274.14432) [16], peak 25 as kirkine (C_16_H_20_O_3_N^+^, 274.14822) [16], and peak 26 as albomaculine (C_19_H_24_O_5_N^+^, 346.17456) [33].

Peak 27 with a [M+H]^+^ ion at *m/z* = 330.14191 was identified as 3-epimacronine also found on *P. australis* [22], peak 28 as 10-*O*-methylpseudolycorine, peak 29 as the isoquinoline alkaloids aknadicine detected on bulbs of *Narcissus tazetta* [34] and peak 30 as 3-*O*-acetylnarcissidine [35]. Peak 31 was identified as 10-*O*-dimethylgalanthine [16], peak 32 as 7-methoxy-*O*-methyllycorenine previously isolated from Brazilian *Hippeastrum aulicum* [28], peak 36 was identified as homolycorine [36], and peak 41 was identified as 5-methyl-2-epihippamine, peak 44 as 3-*O*-methylnarcissidine [37], peak 46 as 2-*O*-acetyl-4-*O*-methyllicorine [16], peak 47 as jonquailine [38], and peak 50 as the classical isoquinoline bulbocapnine. These alkaloids have been isolated previously from *Galanthus nivalis* [39] and detected recently from bulbs of *Narcissus tazetta* [34]. Peak 53 was identified as nerinine detected in some Chilean *Rhodophiala* species [12], peak 55 was identified as 1-*O*-acetylcaranine [40], and peak 58 was identified as the belladine-type alkaloid carltonine A isolated from the bulbs of *Narcissus pseudonarcissus* cv. Carlton [41]. Peak 63 with a [M+H]^+^ ion at *m/z* = 300.12955 was identified as 11-oxo-haemanthamine [28], peak 65 as 1-*O*-acetylbuphanamine isolated from the bulbs of *Boophone disticha* [42], and peak 66 as 11-acetylambelline [43]. Peak 67 with a [M+H]^+^ ion at *m/z* = 288.12863 was identified as maritidine, and was detected previously from *P. australis* bulbs [22], peaks 68 and 69 were proposed as 9-norpluviine [16], and 5-methylipseudolycorine [16].

### 2.2. Enzyme Inhibition Studies

*Phycella cyrtanthoides* bulbs and leaves alkaloid extracts were evaluated *in vitro* for acetylcholinesterase, butyrylcholinesterase and tyrosinase inhibitory effects (Table 2, expressed as IC_50_ values). Bulbs and leaves were active against AChE and BChE. Bulbs showed the highest inhibitory effects compared to leaves with IC_50_ values for AChE of 4.29 ± 0.04 and for BChE of 18.32 ± 0.03 µg/mL. *P. cyrthantoides* bulbs were more active than Chilean *P. australis* bulbs (IC_50_ = 80.12 ± 1.03 µg/mL) [22], while the Argentinian *P. herbetiana* bulbs showed strong inhibitory activity against AChE (IC_50_ = 1.2 ± 0.12 µg/mL) [23]. No results regarding BChE inhibitory activity have been reported for other *Phycella* species. Conversely, no activity was detected against tyrosinase enzyme on the bulbs and leaves of the alkaloid extract of *P. cyrtanthoides*. These results are in agreement with previous studies using isolated and alkaloid fractions [44], since the presence of phenolic groups has positive effects on the inhibitory activity due to the chelating properties of metals such as copper [45].

### 2.3. Docking Studies

All compounds subjected to docking assays in the *Torpedo californica* acetylcholinesterase (*Tc*AChE) catalytic site and human butyrylcholinesterase (*h*BChE) catalytic site turned out to be the major alkaloids selected from *Phycella cyrtanthoides* bulbs and leaves extracts according to the UHPLC chromatogram (Figure 2). Docking experiments were performed to determine the pharmacological behavior of these main alkaloids, and therefore, their contributions to the cholinesterase inhibitory activities. The best docking binding energies expressed in kcal/mol of each selected compound are shown in Table 3.

#### 2.3.1. Acetylcholinesterase (*Tc*AChE) Docking Results

The binding energies shown in Table 3 indicate that 2-α-methoxy-6-*O*-ethyloduline is the main compound responsible for acetylcholinesterase inhibition, even though all derivatives displayed good binding energies over the enzyme, such as 3-*O*-acetylnarcissidine and 3-hydroxydihydrocaranine, which showed energy descriptors of −8.88 kcal/mol and −8.67 kcal/mol, respectively, turning them into good candidates as acetylcholinesterase inhibitors. These results are consistent with the experimental data that showed both bulbs and leaves extracts demonstrated good half-maximal inhibitory concentration (IC_50_ values were within the same order of magnitude in µg/mL), as depicted in Table 2.

The docking assays indicated that all the main alkaloids that are reported in the present study establish hydrogen bond interactions with acetylcholinesterase. In addition to the hydrogen bond interactions, some alkaloids also establish π–π interactions, T-shaped interactions, and salt bridges. For instance, 3-hydroxydihydrocaranine performs the following interactions with acetylcholinesterase: one π–π interaction between the benzene ring of Phe330 and the 1,3-benzodioxole moiety, and two hydrogen bond interactions (one of which occurs between the Glu199 carboxylate group (–COOH) and one of its secondary alcohols (–OH), while the other hydrogen bond interaction occurs between another secondary –OH and the amino acid Ser200 of the acetylcholinesterase catalytic site), as depicted in Figure 4A. Kirkine, as well as 3-hydroxydihydrocaranine, show one π–π interaction with Trp84, and two hydrogen bond interactions with Ser122 and Glu199, respectively (Figure 4B). The compound 10-norpluviine is arranged in a similar manner compared to Kirkine, both having their phenyl moieties of the 2-methoxyphenol frameworks overlapped between them in the enzyme’s catalytic site, leaving the tertiary amino groups of both compounds in opposite directions. This way, 10-norpluviine cannot perform a π–π interaction with Trp84, but instead still shows a good binding energy since it exhibits two hydrogen bond interactions with Tyr130 and through the carbonyl group (C=O) of Trp84. Additionally, 10-norpluviine exhibits a salt bridge with Asp72, which probably contributes to its binding stabilization into the enzyme catalytic site (Figure 4E). 3-*O*-acetylnarcissidine displays an analogous pose into the acetylcholinesterase catalytic site with Kirkine and 10-norpluviine, but the 1,2-dimethoxybenzene core of 3-*O*-acetylnarcissidine is superimposed with the cycloaliphatic rings of the former compounds, performing two hydrogen bond interactions with Gly118 and Asp72, as well as a π–π interaction with Phe330 (Figure 4F).

10-*O*-dimethylgalanthine showed a good binding energy of −9.38 kcal/mol. Inside the acetylcholinesterase catalytic site, this derivative performs an important salt bridge interaction between the amino group and Asp72 amino acid, but also carries out a hydrogen bond interaction with Ser122, as well as π–π and T-shaped interactions through its aromatic 2-methoxyphenol moiety and the residues Phe330 and Tyr334, respectively. Therefore, these data confirm 10-*O*-dimethylgalanthine as a good candidate for acetylcholinesterase binding and inhibition (Figure 4C).

2-α-methoxy-6-*O*-ethyloduline, which demonstrated to possess the best binding energy of all derivatives, presents a slightly different binding mode in the acetylcholinesterase catalytic site compared to 3-hydroxydihydrocaranine. Notwithstanding, both compounds share the same direction of their 1,3-benzodioxole cores and their tertiary amino groups. However, the difference in the binding orientation of the 2-α-methoxy-6-*O*-ethyloduline allows it to execute two hydrogen bond interactions with Glu199 and Tyr121, as well as a T-shaped interaction with Phe330, suggesting that these features are presumably responsible for the better binding energy profile (Figure 4D).

#### 2.3.2. Butyrylcholinesterase (*h*BuChE) Docking Results

The binding energies from the docking assays of the major selected alkaloids from the *Phycella cyrtanthoides* extracts over butyrylcholinesterase also showed good binding energy profiles (Table 3). The half-maximal inhibitory concentration values (IC_50_) for *P*. *cyrtanthoides* bulbs and *P*. *cyrtanthoides* leaves extracts were 18.32 ± 0.03 µg/mL and 37.70 ± 0.02 µg/mL, respectively, indicating that they effectively inhibit the human butyrylcholinesterase. In this manner, the docking experiments confirm the inhibitory potential mentioned above. Kirkine was the alkaloid that exhibited the best binding energy (−8.33 kcal/mol). Kirkine displays three hydrogen bond interactions, two of which are performed by the hydroxyl group (−OH) of its 2-methoxyphenol moiety, where the hydrogen atom of the –OH interacts with the carboxylate group of Glu197, whereas the oxygen atom of the same –OH interacts with Ser198. The third hydrogen bond interaction occurs between the secondary alcohol (–OH) present in one of the cycloaliphatic rings of Kirkine and the amino acid His438 (Figure 5B). Furthermore, the same 2-methoxyphenol aromatic ring of Kirkine is in charge to perform other two T-shaped interactions with the residues Trp82 and His438, whereupon this alkaloid derivative achieves good stability within the butyrylcholinesterase catalytic site (Figure 5B). In the same way, 10-norpluviine is arranged in a similarly mode into the enzyme pocket; in fact, the same hydrogen bond interactions with Glu197, Ser 198 and His438, as well as the T-shaped interaction with His438 could be perceived (Figure 5E). Nonetheless, this derivative does not perform the T-shaped interaction seen in Kirkine with Trp82, which could explain the lower energy shown for 10-norpluviine (Table 3). 10-*O*-dimethylgalanthine is positioned in an opposite manner relative to Kirkine and 10-norpluviine; therefore, the interactions performed by 10-*O*-dimethylgalanthine are executed with different residues of the catalytic cavity, showing two hydrogen bond interactions with Ala328 and His438, one π-cation interaction between the amino group and Trp82, as a well as a salt bridge with the latter amino group and Glu197 (Figure 5C).

3-hydroxydihydrocaranine carries out three hydrogen bond interactions with the amino acids Asp72, Trp82 and Tyr128. Moreover, 3-hydroxydihydrocaranine shares a relatively common binding pose with 3-*O*-acetylnarcissidine into the butyrylcholinesterase catalytic site. Nonetheless, the fact that these two derivatives show similar, but not identical, orientations results in a common hydrogen bond interaction with the amino acid Asp70. A hydrogen bond interaction through one of the oxygen atoms of the 1,3-benzodioxole framework in the case of 3-hydroxydihydrocaranine, and the same hydrogen bond interaction through one of the oxygen atoms of the 1,2-dimethoxybenzene core of 3-*O*-acetylnarcissidine. Likewise, 3-*O*-acetylnarcissidine also carries out another hydrogen bond interaction with Ser198 and the only methoxy group (–OCH_3_) of its structure, which is present in one of its cycloaliphatic rings (Figure 5A,F).

The 2-α-methoxy-6-*O*-ethyloduline established pose into the butyrylcholinesterase catalytic site is quite different relative to the other major alkaloids studied; however, this derivative is still stabilized through two hydrogen bond interactions with Gly116 and Ser198, as well as two T-shaped interactions performed by the 1,3-benzodioxole moiety and the amino acids Trp82 and His438 (Figure 5D).

### 2.4. Antiproliferative Effects

The antiproliferative effects of bulbs and leaves of *P. cyrtanthoides* alkaloid extracts were tested in the following six tumor cell lines: A549 (lung), HBL-100 (breast), HeLa (cervix), SW1573 (lung), T-47D (breast) and WiDr (colon). To the best of our knowledge, no previous studies regarding the antiproliferative potential have been conducted on *Phycella* genera. The alkaloid extracts showed activity against all tumor cell lines in this study. Both extracts display a GI_50_ (50% growth inhibition) <2.5 µg/mL against all cell lines. These results indicate that the potency of the compounds present in the extract is comparable to standard anticancer drugs. For instance, cisplatin under the same six cell lines displayed GI_50_ values in the range 0.54–6.9 µg/mL (Appendix A). Additionally, several alkaloids contained in the extracts have been previously identified to have anticancer activity. Lycorine showed significant antiproliferative effects against A2780 and MV4-11 cells [46]. Previous studies have demonstrated that lycorine and haemanthamine were able to inhibit cell proliferation using a panel of 16 tumor cell lines [5]. In a previous report, some alkaloids, such as norpluviine, caranine, dihydrolycorine, pseudolycorine, and lycorine, were also investigated against A549, OE21, Hs683, U373, SKMEL, and B16F10 cancer cells lines [47]. In addition, from *Hippeastrum solandriflorum*, several isolated alkaloids, including narcissidine, 11-hydroxyvittatine, narciclasine, among others, were evaluated against HCT-116 (colon adenocarcinoma), HL-60 (leukemia), OVCAR-8 (ovarian carcinoma) and SF-295 (glioblastoma) cancer cell lines [48]. On the other hand, the homolycorine-type alkaloid hippeastrine inhibited the proliferation of Hep G2 and HT-29 cells [27]. In addition, albomaculine have been evaluated against breast (Hs578T, MDA-MB-231, MCF7), colon (HCT-15), melanoma (SK-MEL-28) and lung (A549) cells lines [33]. Lycorine and homolycorine-type alkaloids were the most common alkaloids identified in *P. cyrthanthoides,* which could be strongly associated as the main compounds responsible for the antiproliferative effects.

## 3. Materials and Methods

### 3.1. Chemicals

Ultra-pure water (<5 µg/L TOC) was obtained from a water system of purification (Milli-Q Merck Millipore, Chile). Methanol (HPLC grade) and formic acid (puriss. p.a. for mass spectrometry) from J. T. Baker (Phillipsburg, NJ, USA). Acetonitrile (HPLC grade) was from Merck (Santiago, Chile). 2,2-diphenyl-1-picrylhydrazyl (DPPH), gallic acid, DMSO, NaCl, MgCl_2_, acetyl-thiocholine iodide (ATCI), butyryl-thiocholine chloride (BTCl), 5,5′-dithiobis (2-nitrobenzoic acid) (DTNB), sulforhodamine B (SRB), galanthamine, acetylcholinesterase (AChE), butyrylcholinesterase (BChE), and tyrosinase were purchased from Sigma-Aldrich Chem. Co. (St Louis, MO, USA). Lycorine hydrochloride was purchased from Sigma-Aldrich Chem. Co. (St Louis, MO, USA). Fetal calf serum (FCS) was purchased from Gibco (Grand Island, NY, USA). Trichloroacetic acid (TCA), glutamine, and gentamicin were purchased from Merck (Darmstadt, Germany).

### 3.2. Plant Material

*Phycella cyrtanthoides* was collected during the flowering state in the locality of Cachagua, Región de Valparaíso, Chile, in November of 2019 (22°34′26.7″ S, 68°01′24.4″ W). A voucher herbarium specimen (voucher number PC-52019) was deposited in the Laboratory of Natural Products of the Universidad Austral de Chile (Chile). The sample was authenticated by the botanist Jorge Macaya, University of Chile, Santiago, Chile. The entire plant was cleaned and separated into the different organs (Figure 6), dried, and stored without light, and then ground using an electric processor (Ursus Trotter, UT-PETRUS320) to prepare the extracts.

### 3.3. Extraction

Bulbs and leaves (100 g) were extracted three times with 100 mL MeOH using an ultrasonic water bath (UC-60A Biobase, Guanzhou, China) with a procedure similar to that usually reported to Amaryllidaceae alkaloids [49] with some modifications. Briefly, extracts were combined, filtered, and concentrated under reduced pressure. The raw extracts were acidified to pH 2 with H_2_SO_4_ (2% *v*/*v*) and extracted with Et_2_O (3 × 30 mL). The aqueous solutions were basified with 25% NH_3_·H_2_O, up to pH 10. The alkaloids were extracted with EtOAc (3 × 50 mL). The organic layer was evaporated under reduced pressure to obtain the alkaloid extracts.

### 3.4. UHPLC–DAD–MS Instrument

The untargeted analysis of the alkaloid extracts was carried out using a UHPLC-high-resolution MS machine (Thermo Dionex Ultimate 3000 system with DAD detector controlled by Chromeleon 7.2 software hyphenated with a Thermo QExactive MS focus) operated in positive mode [50]. For the analysis, 5 mg of each partition extract were dissolved in 2 mL of methanol, filtered through a 200 µm polytetrafluoroethylene filter, and 10 µL were injected into the instrument. Data acquired were finally analyzed by Xcalibur 2.3 (Thermo Fisher).

### 3.5. LC Parameters and MS Parameters

Liquid chromatography was performed using a UHPLC C18 column (Acclaim, 150 × 4.6 mm ID, 2.5 µm; Thermo Fisher Scientific, Bremen, Germany) operated at 25 °C. The detection wavelengths were 280, 254, 330, and 354 nm, and photodiode array detector was set from 200 nm to 800 nm. A total of 1% formic acid aqueous solution was used as the mobile phase A, and the mobile phase B was acetonitrile. The gradient program was as follows: (0.00 min, 5% B); (5.00 min, 5% B); (10.00 min, 30% B); (15.00 min, 30% B); (20.00 min, 70% B); (25.00 min, 70% B); 35.00 min, 5% B) and 12 min for column prior to each injection equilibration before injections. The flow rate was 1.00 mL/min and the injection volume was 10 µL. Briefly, the parameters are as follows: sheath gas flow rate, 75 units; auxiliary gas unit flow rate, 20; capillary temperature, 400 °C; auxiliary gas heater temperature, 500 °C; spray voltage, 2500 V (for ESI); and S lens, RF level 30. Full scan data in positive mode were acquired at a resolving power of 70,000 FWHM at *m*/*z* 200. The mass scan range was between *m*/*z* 130–1000; automatic gain control (AGC) was set at 3 × 10^6^ and the injection time of 200 ms. The chromatographic system was coupled to MS with a source II heated electro-nebulization ionization probe (HESI II). The nitrogen gas carrier (purity > 99.999%) was obtained from a Genius NM32LA (Peak Scientific, Billerica, MA, USA) generator and used as a collision and damping gas. Mass calibration for Orbitrap and HESI parameters were described previously [45].

### 3.6. Determination of Cholinesterase Inhibition

The inhibitory activity of *P. cyrtanthoides* extracts was evaluated by utilizing the Ellman’s method as previously reported [51,52]. A sample solution (50 µL, 2 mg/mL) was mixed with 120 µL of 5,5-dithio-bis (2-nitrobenzoic) acid (DTNB) 0.3 mM, and AChE (0.26 U/mL, acetylcholinesterase from Electric eel), or BChE (0.26 U/mL, butyrylcholinesterase from horse serum) solution (25 µL) in Tris-HCl buffer 50 mM (pH = 8.0) in a 96-well microplate and incubated for 20 min at 37 °C. The reaction was initiated by the addition of 25 µL of acetylthiocholine iodide (ATCI) 1.5 mM or butyrylthiocholine chloride (BTCl) 1.5 mM. A blank was prepared to all reaction reagents without enzymes solution. The absorbances were recorded at three times at 405 nm during 30 min at 37 °C using microplate reader (Synergy HTX Multi-Mode). Galanthamine hydrobromide was used as a positive control. The cholinesterase inhibitory activity was expressed as IC_50_ (μg/mL, concentration range 0.5 to 50 μg/mL). All data were recorded in triplicate.

### 3.7. Determination of Tyrosinase Inhibition

Tyrosinase inhibitory activity was evaluated by utilizing the dopachrome method as previously reported [45]. *P. cyrtanthoides* extract solution (20 µL, 2 mg/mL) was mixed with Mushroom tyrosinase solution (100 unit/mL, 40 µL) and phosphate buffer 0.067 M (30 µL, pH = 6.8) in a 96-well microplate and incubated for 15 min at 30 °C. The reaction was initiated with the addition of 40 µL L-DOPA 2.5 mM and the mixture was incubated for 15 min at 25 °C. A blank was prepared to all reaction reagents without enzyme. The sample and blank absorbances were recorded at 492 nm using a microplate reader (Synergy TM HT Multi-Mode). Kojic acid was used as a positive control. The tyrosinase inhibitory activity was expressed as IC_50_ (μg/mL, concentration range 31.25 to 250 μg/mL). All data were recorded in triplicate.

### 3.8. Docking Assays

Docking simulations were carried out for selected major alkaloids, shown in Appendix A, obtained from *Phycella cyrtanthoides* leaves or bulbs extracts. The geometries and partial charges of each alkaloid were fully optimized using the DFT/B3LYP method with standard basis set 6-311G/+dp [53,54] in Gaussian 09W software. Crystallographic enzyme structures of *Torpedo Californica* acetylcholinesterase (*Tc*AChE; PDBID: 1DX6 code) [55], and human butyrylcholinesterase (*h*BuChE; PDBID: 4BDS code) [56] were downloaded from the Protein Data Bank RCSB PDB [57] (for full description, see Appendix A).

### 3.9. Antiproliferative Activity

Antiproliferative activity was evaluated using human solid tumor cell lines. Cells were inoculated onto 96-well plates using 100 μL per well at densities of 2500 (A549, HBL-100, and HeLa) and 5000 (SW1573, T-47D, and WiDr) cells per well. Extract solutions dispersed in water were dissolved in DMSO at 400 times the final maximum test concentration (250 µg/mL). Control cells were exposed to an equivalent concentration of DMSO (0.25% *v*/*v*, negative control). The extracts were tested in triplicate at concentrations ranging from 250 to 2.5 µg/mL. Treatment with compounds started on day 1 after plating. Incubation time with compounds was 48 h, after which cells were precipitated with ice-cold trichloroacetic acid (TCA) (50% *w*/*v*, 25 μL) during 60 min at 4 °C. Then, the sulforhodamine B (SRB) assay was performed. The optical density (OD) was measured at 530 nm using BioTek PowerWave XS microplate reader. The results were expressed as GI_50_ values (µg /mL, calculated according to NCI formulas).

### 3.10. Statistical Analysis

The results obtained from these experiments were repeated five times and expressed as mean ± standard error of mean. Statistical analysis of the data was performed using analysis of variance (two-way ANOVA) where applicable followed by post hoc Bonferroni test. In addition, the determination of the sensitivity (EC_50_ or IC_50_) was performed using nonlinear regression (sigmoidal) via origin Pro 9.0 software package (Origin lab Corporation, Northampton, MA, USA). Statistical significance was set at *p* < 0.05.

## 4. Conclusions

In summary, seventy alkaloids were detected in bulbs and leaves from the endemic Amaryllidaceae plant *Phycella cyrtanthoides* using UHPLC-DAD-Orbitrap-MS mass spectrometry analysis. Lycorine and haemanthamine type were the major alkaloids identified. The alkaloids extract showed activity against AChE and BChE, but no activity was observed against tyrosinase enzymes. Bulbs’ extracts proved to be the most active against both cholinesterase enzymes. The docking results indicated that hydrogen bond and T-shaped interactions are responsible for better binding over AChE and BChE enzymes. The assessment of the antiproliferative activity indicates that bulbs and leaves exert activity against several human tumor cell lines. The results reported herein are promising and indicate that alkaloid compounds that are present in *Phycella cyrtanthoides* extracts should be further studied for their antiproliferative activities as well as their potential therapeutic applications against neurodegenerative diseases. However, the isolation of major alkaloids as well as *in vivo* studies are needed to further evaluate the pharmacological properties of this plant.

## Figures and Tables

**Figure 1 metabolites-12-00188-f001:**
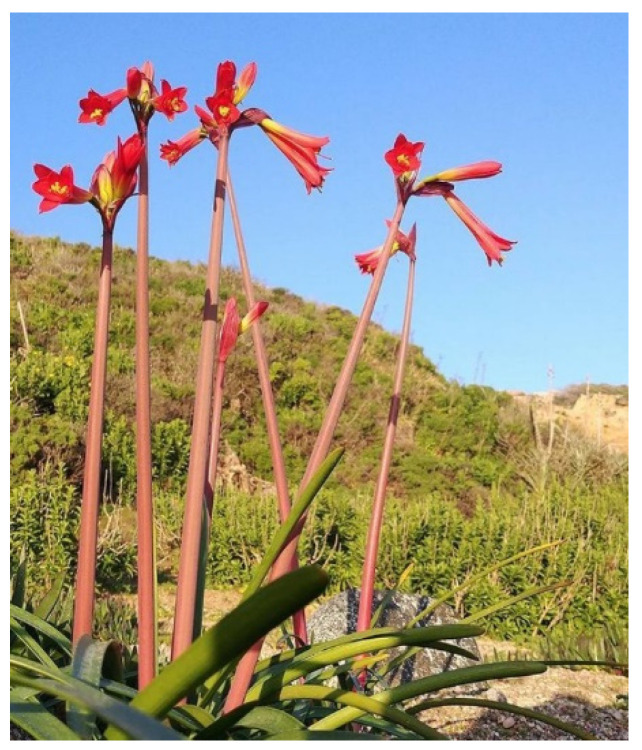
*Phycella cyrtanthoides* (Amaryllidaceae) plants.

**Figure 2 metabolites-12-00188-f002:**
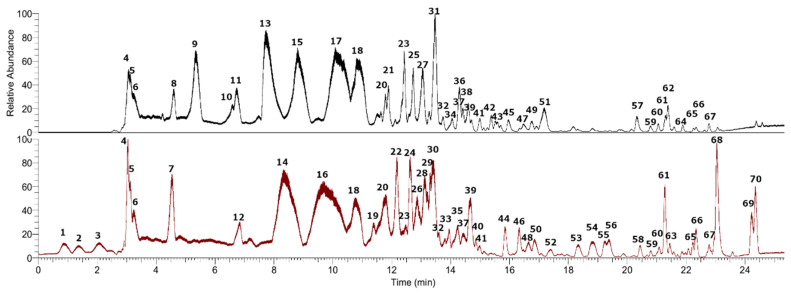
*Phycella cyrtanthoides*: entire plant (**left**), bulbs (**middle**), and leaves (**right**).

**Figure 3 metabolites-12-00188-f003:**
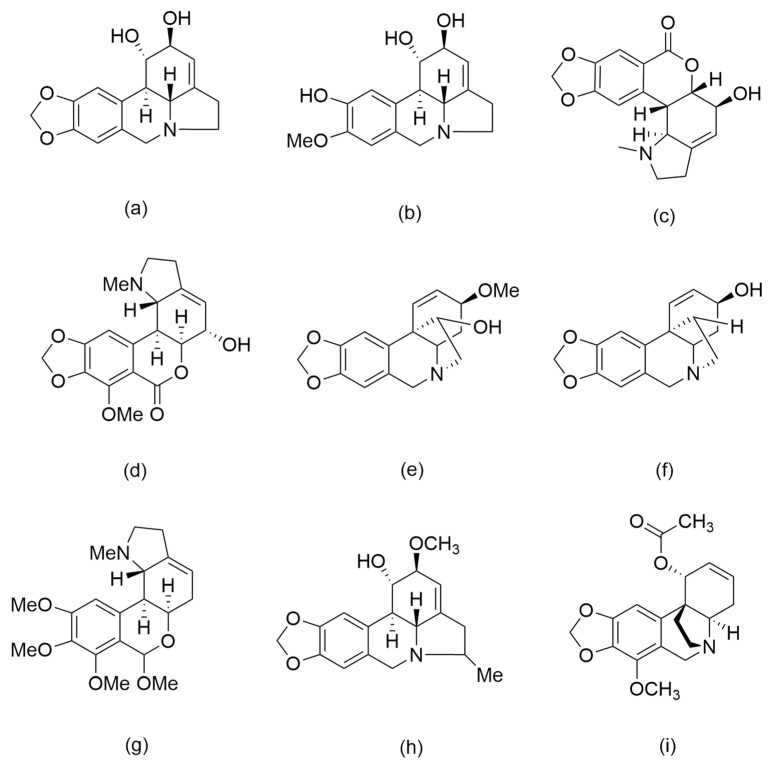
UHPLC chromatogram of *Phycella cyrtanthoides* bulbs (black) and leaves (red) in positive mode.

**Figure 4 metabolites-12-00188-f004:**
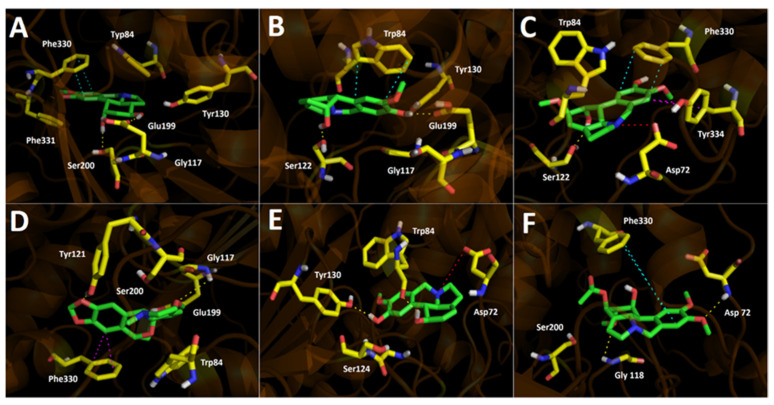
Lycorine (**a**), pseudolycorine (**b**), hippeastrine (**c**), candimine (**d**), haemanthamine (**e**), vittatine (**f**), 7-methoxy-*O*-methyllycorenine (**g**), 5-methyl-2-epihippamine (**h**) and 1-*O*-acetylbuphanamine (**i**).

**Figure 5 metabolites-12-00188-f005:**
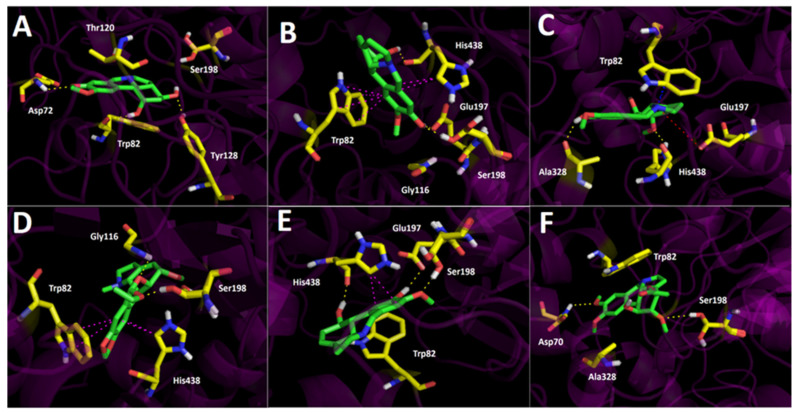
Predicted binding mode and predicted intermolecular interactions of major alkaloids in leaves and bulbs of *Phycella cyrtanthoides* extracts and the residues of *Torpedo californica* acetylcholinesterase (*Tc*AChE) catalytic site; (**A**) 3-hydroxydihydrocaranine in the catalytic site; (**B**) Kirkine in the catalytic site; (**C**) 10-*O*-dimethylgalanthine in the catalytic site; (**D**) 2-α-methoxy-6-*O*-ethyloduline in the catalytic site; (**E**) 10-norpluviine in the catalytic site; (**F**) 3-*O*-acetylnarcissidine in the catalytic site. Yellow dotted lines indicate hydrogen bond interactions; cyan dotted lines represent π–π interactions; magenta dotted lines represent T-shaped interactions; and red dotted lines indicate salt bridge interactions.

**Figure 6 metabolites-12-00188-f006:**
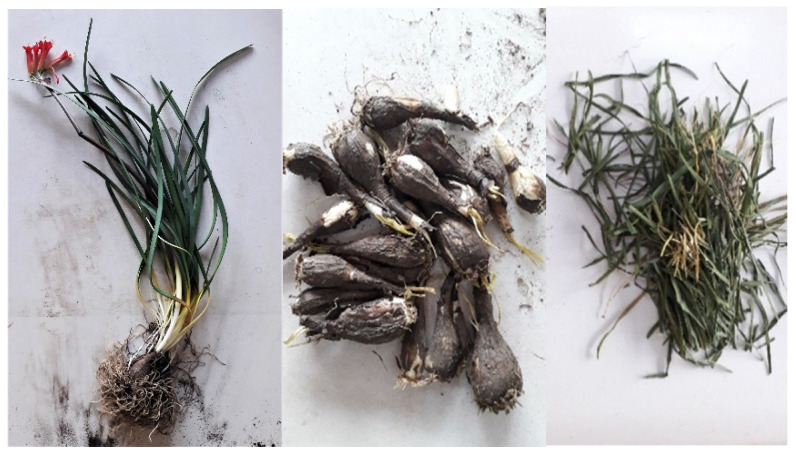
Predicted binding mode and predicted intermolecular interactions of major alkaloids in leaves and bulbs *Phycella cyrtanthoides* extracts and the residues of the human butyrylcholinesterase (*h*BuChE) catalytic site; (**A**) 3-hydroxydihydrocaranine in the catalytic site; (**B**) Kirkine in the catalytic site; (**C**) 10-*O*-dimethylgalanthine in the catalytic site; (**D**) 2-α-methoxy-6-*O*-ethyloduline in the catalytic site; (**E**) 10-norpluviine in the catalytic site; (**F**) 3-*O*-acetylnarcissidine in the catalytic site. Yellow dotted lines indicate hydrogen bond interactions; cyan dotted lines represent π–π interactions; magenta dotted lines represent T-shaped interactions; blue dotted lines indicate π-cation interactions; and red dotted lines indicate salt bridge interactions.

**Table 1 metabolites-12-00188-t001:** Ultrahigh performance liquid chromatography orbitrap mass spectrometry (UHPLC-Orbitrap-MS) identification of *Phycella cyrtanthoides*.

Peak	UV Max	Tentative Identification Name (AA-Type)	Elemental Composition [M+H] ^+^	Rt	Theoretical Mass (*m/z*)	Measured Mass (*m/z*)	Accuracy (ppm)	MSn Ions (ppm)	Organs
1	236–281	2-Hydroxyalbomaculine (homolycorine-type)	C_19_H_24_O_6_N^+^	0.86	362.15981	362.13348	−72.719	274.14774, 251.09735, 214.06267, 199.07582, 165.07063, 121.06544, 107.04959	L
2	231–248-271–307	Tazettine isomer (tazettine-type)	C_18_H_22_O_5_N^+^	1.37	332.14925	332.15756	25.020	316.12589, 274.14835, 247.12236, 228.14095, 181.06543, 152.06271, 115.05478, 107.04965	L
3	233–247-306	Powelline (crinine-type)	C_17_H_20_O_4_N^+^	2.06	302.13868	302.14523	21.663	284.13364, 274.14798, 251.09767, 228.06763, 181.06557, 165.07068, 127.05473, 115.05476,	L
4	236–286	Lycorine * (lycorine-type)	C_16_H_18_O_4_N^+^	3.07	288.12303	288.12869	19.628	286.11295, 239.05540, 194.11844, 166.12335, 147.04440, 119.04981, 103.05488	B; L
5	235–283	Pseudolycorine (lycorine-type)	C_16_H_20_O_4_N^+^	3.11	290.13923	290.14432	19.423	244.09990, 214.08778, 177.26830, 153.07031, 152.06248, 147.04427, 119.04977, 112.91312	B; L
6	233–278	Hippeastrine isomers (homolycorine-type)	C_17_H_18_O_5_N^+^	3.25	316.11850	316.12579	25.689	298.11389, 290.14404, 280.10153, 274.14795, 191.03445, 166.12321, 121.06518	B; L
7	231–276	Hippeastrine isomers (homolycorine-type)	C_17_H_18_O_5_N^+^	4.54	316.11850	316.12579	24.803	290.14407, 274.14798, 191.03456, 166.12329, 121.06531	L
8	232–278	Hippeastrine isomers (homolycorine-type)	C_17_H_18_O_5_N^+^	4.59	316.11850	316.12616	25.973	291.14764, 274.14828, 191.03468, 247.12253, 166.12341, 121.06541	B
9	233–289	Pluviine (lycorine- type)	C_17_H_22_O_3_N^+^	5.35	288.15942	288.12869	19.628	275.25861, 270.11700, 240.06802, 194.09659, 165.07004, 147.04448, 119.04977	B
10	232–289	Dihydrolycorine (lycorine- type)	C_16_H_20_O_4_N^+^	6.59	290.13923	290.14432	19.423	272.13245, 257.15308, 220.12132, 167.11861, 152.06258, 149.06068	B
11	233–290	3-*O*-methyl-epimacowine (crinine-type)	C_16_H_18_O_4_N^+^	6.73	288.12358	288.12872	19.732	193.65796, 179.76768, 153.07051, 128.55969, 115.05492, 109.22836	B
12	231–277-310	11-*O*-methylcrinamine (crinine-type)	C_18_H_22_O_4_N^+^	6.84	316.15433	316.12589	−89.979	631.24774 (2M^+^), 274.14807, 228.14088, 182.11839, 121.06535	L
13	231–277	3-Hydroxydihydrocaranine (lycorine- type)	C_16_H_20_O_4_N^+^	7.74	290.13868	290.14420	19.940	268.67581, 239.88618, 167.34416, 137.10796, 111.88570, 107.48717	B
14	232–278	*epi*-Zephyranthine isomers (lycorine- type)	C_16_H_20_O_4_N^+^	8.34	290.13868	290.14413	18.768	272.13232, 262.11163, 244.09996, 214.08743, 181.06534, 169.06546, 147.04451, 120.08131, 118.06568	L
15	232–278	*epi*-Zephyranthine isomers (lycorine- type)	C_16_H_20_O_4_N^+^	8.81	290.13868	290.14435	19.526	272.13245, 268.74606, 181.17052, 170.64424, 148.16725, 129.39676, 118.06547	B
16	232–275	Hippeastrine isomers (homolycorine-type)	C_17_H_18_O_5_N^+^	9.67	316.11850	316.12579	24.80	298.11267, 290.14447, 274.14795, 191.03447, 166.12320, 121.06516	L
17	232–275	Hippeastrine isomers (homolycorine-type)	C_17_H_18_O_5_N^+^	10.09	316.11850	316.12610	−22.476	288.12851, 274.14813, 191.03467, 166.12337, 137.10789, 124.07613	B
18	235–282-314	Candimine (homolycorine-type)	C_18_H_20_O_6_N^+^	10.75	346.12906	346.13831	28.301	316.12576, 288.12836, 274.14810, 228.14084, 155.15477, 138.05533, 121.06532	B; L
19	234–385	8-*O*-demethylmaritidine (haemanthamine-type)	C_16_H_20_O_3_N^+^	11.39	274.14377	274.14804	15.576	267.06967, 223.07715, 191.03459, 177.01894, 149.02383, 121.06531, 107.04959	L
20	232–273	Haemanthamine (haemanthamine-type)	C_17_H_20_O_4_N^+^	11.78	302.13923	302.14548	22.490	289.13168, 272.13251, 228.14088, 183.57773, 180.10258, 167.11855, 161.10791, 144.08138	B; L
21	235–282	5-Methyl-epimethylpseudolycorine (lycorine-type)	C_18_H_24_O_4_N^+^	11.89	318.17053	318.17819	25.788	287.12823, 162.06857, 147.04459, 125.98681, 115.05488, 103.05471	B
22	241–325	2α-Methoxy-6-*O*-ethyloduline (homolycorine-type)	C_20_H_26_O_5_N^+^	12.16	360.18110	360.19092	28.792	330.14169, 274.14807, 270.11676, 228.14082, 153.10274, 151.07596, 121.06521	L
23	240–291	Vittatine (haemanthamine- type)	C_16_H_18_O_3_N^+^	12.43	272.12867	272.13257	16.242	268.10175, 247.12251, 199.21875, 180.10249, 167.99812, 153.13918, 121.06526, 115.05503	B; L
24	233–288	10-Norpluviine (lycorine-tipe)	C_16_H_20_O_3_N^+^	12.63	274.14432	274.14813	15.904	274.14813, 256.13666, 228.14076, 175.03946, 147.04443, 121.06533, 118.06563, 102.03407	L
25	236–287	Kirkine (lycorine-tipe)	C_16_H_20_O_3_N^+^	12.73	274.14432	274.14822	16.232	270.11679, 256.13672, 231.15173, 228.07106, 197.16562, 175.03972, 120.08125, 118.06559	B
26	232–272	Albomaculine (homolycorine-type)	C_19_H_24_O_5_N^+^	12.87	346.16490	346.17456	27.994	320.15652, 304.16083, 274.14795, 193.05043, 180.10242, 178.06317, 152.06268, 103.05467	L
27	232–256-309	3-Epimacronine (tazettine-type)	C_18_H_20_O_5_N^+^	13.05	330.13415	330.14191	25.173	326.09451, 316.12592, 247.12227, 231.15181, 202.13469, 167.15497, 144.08133, 111.09221	B
28	234–252	10-*O*-methylpseudolycorine (lycorine-type)	C_17_H_22_O_4_N^+^	13.12	304.15488	304.16098	21.848	304.16074, 276.12741, 258.11655, 193.05020, 178.06328, 165.07066, 147.04482, 125.08409, 118.06564	L
29	232–271	Aknadicine	C_19_H_24_O_5_N^+^	13.29	346.16545	346.17526	29.929	316.12640, 304.16125, 193.05049, 178.06313, 125.08411, 121.06541, 110.06061	L
30	248–271	3-*O*-Acetylnarcissidine (lycorine-type)	C_20_H_26_O_6_N^+^	13.43	376.17601	376.18796	33.217	316.12589, 304.16135, 258.11682, 193.05042, 165.07048, 153.07048, 147.04456, 125.08416, 118.06568	L
31	238–284	10-*O*-Dimethylgalanthine (lycorine-type)	C_17_H_22_O_4_N^+^	13.48	304.15488	304.16119	22.539	274.14847, 266.08557, 258.11697, 191.15506, 167.15486, 125.98666, 118.06574	B
32	248–271	7-Methoxy-*O*-methyllycorenine (homolycorine-type)	C_20_H_28_O_5_N^+^	13.75	362.19675	362.20648	29.405	346.13885, 330.14188, 247.12267, 221.16689, 191.15508, 167.01363	B; L
33	238–271	Unknown alkaloid	C_24_H_26_O_4_N^+^	13.97	392.18618	392.18298	−6.769	376.18741, 344.15909, 304.16098, 252.10530, 212.14474, 180.10254	L
34	249–278	Unknown alkaloid	C_20_H_24_O_4_N^+^	14.05	342.17053	342.17990	28.977	337.19955, 316.12595, 259.18481, 247.12242, 194.11845, 144.08136	B
35	241–287	Unknown alkaloid	C_19_H_22_O_5_N^+^	14.24	344.16183	344.15912	28.681	337.19943, 316.16238, 282.11783, 227.08345, 191.14377, 110.02042	L
36	248–284	Homolycorine (homolycorine-type)	C_18_H_22_O_4_N^+^	14.29	316.15488	316.16257	−22.206	312.16776, 284.18124, 272.13272, 251.15782, 201.13953, 181.17062, 125.98682, 110.02058	B
37	249–284	Unknown alkaloid	C_14_H_30_O_9_N^+^	14.42	356.19151	356.19595	12.471	226.28473, 201.05020, 143.05002, 115.05481, 108.08139	B; L
38	249–282	Unknown alkaloid	C_18_H_34_O_7_N^+^	14.60	376.23298	376.22391	−24.105	356.19583, 322.15192, 240.15221, 181.17058, 167.01340, 125.98679	B
39	242	Unknown alkaloid	C_18_H_34_O_8_N^+^	14.68	392.22789	392.21906	−22.522	374.17166, 346.17444, 290.15924, 197.11798, 150.09190, 121.06519	B; L
40	243	Unknown alkaloid	C_24_H_24_O_5_N^+^	14.85	406.16490	406.16412	−7.395	392.21921, 346.10187, 290.15942, 274.14810, 211.17101, 197.11813, 179.10728	L
41	253–282	5-methyl-2-epihippamine isomers (lycorine-type)	C_18_H_22_O_4_N^+^	14.99	316.15433	316.12613	25.878	284.18137, 272.13257, 254.16881, 247.12253, 197.08179, 144.08141, 125.98682, 111.09224	B; L
42	252–276	Unknown alkaloid	C_19_H_30_O_6_N^+^	15.37	368.20676	368.19748	32.171	316.12601, 249.14214, 228.14104, 209.20259, 167.01352, 110.02060	B
43	252–278	Unknown alkaloid	C_28_H_33_O_5_N^+^	15.53	463.23532	463.24121	12.705	449.26157, 431.21402, 346.13846, 249.14217, 225.12744, 163.07600	B
44	251–301	3-*O*-methylnarcissidine (lycorine-type)	C_19_H_26_O_5_N^+^	15.85	348.18055	348.19052	28.636	346.17471, 274.14804, 197.11795, 171.14987, 138.09195, 121.06521	L
45	250–280	Unknown alkaloid	C_26_H_32_O_3_N^+^	15.96	406.23767	406.23715	−1.281	346.13843, 316.12604, 247.12244, 203.11884, 144.08141,	B
46	252–297	2-*O*-acetyl-4-*O*-methyllicorine (lycorine-type)	C_19_H_22_O_6_N^+^	16.33	360.14471	360.15427	28.061	314.14539, 267.12448, 247.12241, 211.07701, 171.14987, 121.06534, 103.05473	L
47	253–283	Jonquailine (tazettine-type)	C_19_H_24_O_5_N^+^	16.45	346.16490	346.13846	−76.384	321.20441, 314.14566, 288.05582, 247.12248, 171.14989, 102.03439	B
48	261–305	Unknown alkaloid	C_19_H_22_O_6_N^+^	16.63	360.14416	360.15417	27.783	344.12250, 326.14676, 316.12579, 304.16077, 247.12231, 180.10242, 164.10765	L
49	252–282	Unknown alkaloid	C_27_H_33_O_3_N^+^	16.76	419.24550	419.24805	6.093	398.21057, 346.13849, 316.12604, 268.13763, 210.12157, 121.06532	B
50	250–275	Bulbocapnine (isoquinoline alkaloid)	C_19_H_20_O_4_N^+^	16.85	326.13923	326.14673	24.668	282.08136, 274.14810, 240.13322, 225.15038, 207.13930, 138.09189, 121.06530	L
51	253–278	Unknown alkaloid	C_21_H_32_O_4_N^+^	17.18	362.23258	362.22394	−23.866	360.21735, 316.12598, 247.12244, 167.01346, 102.03439	B
52	254–297	Unknown alkaloid	C_18_H_26_O_7_N^+^	17.39	368.17038	368.17639	16.328	346.17471, 316.12595, 304.16098, 274.14807, 164.10770, 102.03436	L
53	253–302	Nerinine (homolycorine-type)	C_19_H_25_O_5_N^+^	18.34	347.17272	347.17801	59.168	346.17453, 316.12576, 247.14180, 197.11798, 152.10750, 102.03431	L
54	242–277	Unknown alkaloid	C_24_H_16_O_3_N^+^	18.82	366.11247	366.10648	28.525	344.12265, 316.12579, 274.14801, 256.06381, 167.01337, 110.02050, 102.03431	L
55	285–310	1-*O*-acetylcaranine (lycorine-type)	C_18_H_20_O_4_N^+^	19.24	314.13868	314.10910	−94.186	304.16092, 278.08636, 270.15344, 252.10562, 247.12234, 226.18208, 191.03456, 102.03433	L
56	266–309-356	Unknown alkaloid	C_23_H_20_O_3_N^+^	19.41	358.14377	358.13867	−14.240	336.09265, 314.10925, 278.08649, 191.03464, 102.03435	L
57	251–282	Unknown alkaloid	C_14_H_21_O_3_N^+^	20.32	251.15159	251.15793	25.223	250.14595, 228.14110, 186.09221, 167.01358, 102.03442	B
58	253–327	Carltonine A (belladine-type)	C_27_H_33_O_3_N_2_^+^	20.44	433.24857	433.26575	39.654	399.18460, 388.15140, 376.18735, 316.12582, 247.12230 167.01335, 122.54755	L
59	251–278	Unknown alkaloid	C_27_H_38_O_6_N^+^	20.79	472.26936	472.28061	23.811	449.26141, 429.29208, 346.13840, 247.12242 144.08139	B; L
60	245–284	Unknown alkaloid	C_24_H_22_O_4_N^+^	21.06	388.15433	388.15140	−7.561	330.10541, 316.12585 274.14807, 225.55421, 187.12723, 167.01343	B; L
61	251–285	Unknown alkaloid	C_15_H_27_O_3_N^+^	21.27	269.19909	269.20554	25.983	267.19000, 247.12218, 235.17180 184.10028, 150.09189, 121.06524	B; L
62	252–281	Unknown alkaloid	C_20_H_33_O_6_N^+^	21.38	383.23024	383.23532	13.258	352.34933, 316.12595, 269.20566, 228.14091, 221.15578, 144.08133	B
63	252–297	11-Oxo-haemanthamine (haemanthamine-type)	C_17_H_18_O_4_N^+^	21.46	300.12303	300.12955	21.709	277.19623, 269.20547, 240.25401, 239.25075, 211.08781, 180.10248, 102.03432	L
64	253–282	Unknown alkaloid	C_14_H_21_O_3_N^+^	21.89	251.15159	251.15799	25.462	247.12265, 212.14493, 186.12865, 167.01361, 144.08147, 102.03445	B
65	255–302	1-*O*-acetylbuphanamine (crinine-type)	C_19_H_22_O_5_N^+^	22.23	344.14925	344.15903	−14.851	294.11795, 274.14795, 197.11797, 181.12305, 174.12837, 138.09187	B; L
66	257	11-Acetylambelline (crinine-type)	C_20_H_24_O_6_N^+^	22.33	374.16036	374.17181	32.060	358.13858, 324.13116, 304.16089, 197.11806. 174.12848, 151.11229, 121.06525	B; L
67	255–306	Maritidine (haemanthamine type)	C_17_H_22_O_3_N^+^	22.78	288.15942	288.12863	27.502	244.13684, 216.14043, 191.03470, 167.01357, 122.54771, 102.03446	B; L
68	257–296	9-Norpluviine (lycorine-type)	C_16_H_20_O_3_N^+^	23.05	274.14432	274.14798	15.357	274.14798, 256.26694, 230.25032, 228.27116, 191.03436, 174.12823, 147.18108, 121.06519, 102.03426	L
69	258	5-Methylipseudolycorine (lycorine-type)	C_17_H_22_O_4_N^+^	24.22	304.15433	304.16098	21.848	258.28345, 242.28685, 228.27103, 174.12837, 151.11229, 102.03432	L
70	257–292	Unknown alkaloid	C_24_H_26_O_3_N^+^	24.37	376.19072	376.18744	−8.720	352.34924, 274.14795, 258.28336, 174.12845, 166.11258, 146.09694, 132.08127	L

Abbreviations: L = leaves; B = bulbs; RT = retention time; * = identified using authentic compounds.

**Table 2 metabolites-12-00188-t002:** Enzymatic inhibitory activity of *Phycella cyrtanthoides* alkaloid extracts.

Assay	AChE Inhibition IC_50_ (µg/mL)	BChE Inhibition IC_50_ (µg/mL)	Tyrosinase Inhibition IC_50_ (µg/mL)
*P. cyrtanthoides* bulbs	4.29 ± 0.04	18.32 ± 0.03	ND
*P. cyrtanthoides* leaves	8.66 ± 0.03	37.70 ± 0.02	ND
Galanthamine	0.55 ± 0.03	3.82 ± 0.02	-
Kojic acid	-	-	0.76 ± 0.05

All values are expressed as means ± SD (n = 3). Abbreviations: AChE, acetylcholinesterase; BChE, butyrylcholinesterase; ND, not detected (>250 µg/mL).

**Table 3 metabolites-12-00188-t003:** Binding energies obtained from docking experiments of selected major alkaloids in *Phycella cyrtanthoides* bulbs and leaves extracts, as well as the known inhibitor galanthamine over acetylcholinesterase (*Tc*AChE) and butyrylcholinesterase (*h*BChE).

Compound	Binding Energy (kcal/mol) Acetylcholinesterase	Binding Energy (kcal/mol) Butyrylcholinesterase
3-hydroxydihydrocaranine (13)	−8.67	−8.06
Kirkine (25)	−8.42	−8.33
10-*O*-dimethylgalanthine (31)	−8.27	−7.1
2-α-methoxy-6-*O*-ethyloduline (22)	−9.38	−8.21
10-norpluviine (24)	−8.56	−7.48
3-*O*-acetylnarcissidine (30)	−8.88	−7.68
Galanthamine	−11.81	−9.5

## Data Availability

Data is contained within the article or Appendix A. The raw UHPLC MS data or other additional data presented in this study are available on request from the corresponding author. The raw data are not publicly available due to privacy.

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
