# Peer review of "Alkaloid Profiling, Anti-Enzymatic and Antiproliferative Activity of the Endemic Chilean Amaryllidaceae Phycella cyrtanthoides"

_metabolites, 2022, doi:10.3390/metabo12020188_

Round 1

Reviewer 1 Report

In my opinion manuscript is well prepared and gives new information on metabolites of Phycella cyrtanthides leaves and bulbs; activity of extracts against three enzymes and antiproliferative activity against six tumor cell lines. Material and methods are carefully described. I suggest to add Table 4 with GI50 results (or to supplementary material).  Current sentence in line 426 presents it too briefly.

Author Response

Reply to Reviewers

Responses are in blue color

General - Typos and misspellings encountered

Page 1 line 10: Affiliation of authors A.P. and J.M.P. has been revised and corrected

Page 1 line 28: “µM” was corrected to “µg/mL”

Page 5 line 187: “100 µM” was corrected to “100 µg/mL”

Page 5 line 189: “0.001 to 100 μM” was corrected to “250 to 2.5 µg/mL”

Page 18 line 427: “µM” was corrected to “µg/mL”

Page 19 line 462: Funding of authors A.P. and J.M.P. has been revised and corrected

Reviewer 1

Yes

Can be improved

Must be improved

Not applicable

Does the introduction provide sufficient background and include all relevant references?

(x)

( )

( )

( )

Is the research design appropriate?

(x)

( )

( )

( )

Are the methods adequately described?

(x)

( )

( )

( )

Are the results clearly presented?

( )

(x)

( )

( )

Are the conclusions supported by the results?

(x)

( )

( )

( )

Comments and Suggestions for Authors

In my opinion manuscript is well prepared and gives new information on metabolites of Phycella cyrtanthides leaves and bulbs; activity of extracts against three enzymes and antiproliferative activity against six tumor cell lines. Material and methods are carefully described.

I suggest to add Table 4 with GI50 results (or to supplementary material). 

R: Thank you so much to the detailed revision, we have added Table (Table S1) as suggested in supplementary material

Current sentence in line 426 presents it too briefly.

R: Thanks for the comments. We have improved and explained from line 426-430.

Reviewer 2

Yes        Can be improved Must be improved              Not applicable

Does the introduction provide sufficient background and include all relevant references?

(x)          ( )           ( )           ( )

Is the research design appropriate?

(x)          ( )           ( )           ( )

Are the methods adequately described?

( )           (x)          ( )           ( )

Are the results clearly presented?

(x)          ( )           ( )           ( )

Are the conclusions supported by the results?

(x)          ( )           ( )           ( )

Comments and Suggestions for Authors

The manuscript entitled «Alkaloid Profiling, Anti-Enzymatic and Antiproliferative Activity of

The Endemic Chilean Amaryllidaceae: Phycella cyrtanthoides» is an orginal study realized on bulbs and leaves alkaloid extracts of this plant, analyzed using ultrahigh performance liquid chromatography orbitrap mass spectrometry (UHPLC-Orbitrap-MS) and tested for biological activities : enzyme inhibition potential against cholinesterases (acetylcholinesterase, butyrylcholinesterase and tyrosinase ), and antiproliferative effects against tumor cell lines (human cervical (HeLa), lung (A549, SW1573), colon (WiDr), and breast (HBL-100, T-47D). The subject was well-documented and experiment design was smartly conceived and perfectly performed and so yielding convincing conclusions. But minor revision is required to improve the manuscript. below are a few comments and concerns:

Some spelling errors troughout the manuscript should be corrected by carefully checking the whole text, below some examples of errors on words to put in italics:

- Pg 15, Ln 280 : Please put in italic « in vitro »

R: Fixed

- Pg 15, Ln 280, Ln 284, Ln 289 : Please put in italic « Phycella. cyrtanthoides»

R: Fixed

- Pg 15, Ln 286 : Please put in italic « P. herbetiana »

R: Fixed

- Pg 15, Ln 287 : Please put in italic « Phycella »

R: Fixed

So, I recommend the acceptation of this article « Metabolites » journal after these minor required revision.

R: Thanks for the comments. We have improved the article regarding your revisions.

Reviewer 3

Dear authors,

The manuscript entitled "Alkaloid Profiling, Anti-Enzymatic and Antiproliferative Activity of The Endemic Chilean Amaryllidaceae: Phycella cyrtanthoides” was to investigate the alkaloid profiling from the endemic Phycella cyrtanthoides from bulbs and leaves organs. It presents scientific relevance for the area of Plant Science and Medicine area.

After consulting www.sciencedirect.com and https://pubmed.ncbi.nlm.nih.gov/, publications were found for some authors involving the theme. However, you need to change some details/information in the Introduction, Material and Methods, Results, discussion and “conclusions”.

Introduction - It is well written, but I suggest:

-line 68: Remove double space.

R: changed

-Some paragraphs are too short! I suggest joining them, for better understanding and continuity of ideas.

R: Thanks for the revision. We have improved the introduction according to your comment.

- As the article involves, also, the development and validation of the analytical method, I suggest including information on analytical techniques and more references on the determination of Alkaloid profiling and in biological samples (plants in general, or for the reported species) by UHPLC–DAD–MS.

R: We have improved the introduction according to your comment from line 50 to 56.

-I suggest at the end of the introduction, I suggest highlighting the "innovative" proposal of the method, as well as the advantages/disadvantages.

R: Thanks for the comment. We have improved according to your comment.

Plant Material

- Page 3, “Protocol for obtaining extracts” section: Was this protocol based on any references? If yes, add!

R: The extraction process is very well-Known acid-base extraction methods were applied to obtain the rich alkaloids extracts. The reference was added.

-line 132, 140, 158, 155, 166, 168: °C

R: Fixed

-line 174: Remove yellow mark

R: Fixed

Results section:

-page 278: Add abbreviations L, B according to table 1.

R: Thanks for the revision. We have added the missing information

- line 175, 280, 293, 360, 412: to write “Phycella cyrtanthoides” in italic and, throughout the manuscript!

R: Fixed

-line 342: .

R: Added

- Table 3: why the values are marked with "-".

R: It is a negative value

Conclusion: Adequate, but I suggest to indicate disadvantages/limitations of the method and the study! Perhaps, to highlight the text in the 'Limits of the study' section.

R: We have improved the conclusion according to the comment.

Reviewer 4

Comments and Suggestions for Authors

line 57: italicize "Phycella"

R: changed

text in general: galantamine or galanthamine?

R: Thanks for the comments. We have changed to galanthamine in whole text.

text in general (also captions): P. cyrtanthoides and P. herbetiana must be italicized

R: repaired

line 135: change to "as follows"

R:  Fixed

lines 178 and 297 and 360: Torpedo californica

R:  Fixed in all the lines

line 208: some alkaloids? only one (lycorine)

R: changed

lines 241 and 270: change to "peaks"

R: changed

line 262: change to "These alkaloids"

R: changed

lines 315-318: THIS CONCEPT IS TOO SPECULATIVE without a proper quantification of the single components how is it possible to know if there aren't antagonistic effects or if the alkaloids compete for the same binding site?

R: Thanks. We agree with your comments, and we decided to eliminate that paragraph. Quantitation for single components is designed for another manuscript.

docking studies: only CAS in AChE was considered? what about PAS?

R: We gratefully appreciate the reviewer comment, it is an interesting and constructive point. In this sense, we must consider that acetylcholinesterase is a homodimer bound to the plasma membrane. The monomer is an α/β protein with an ellipsoidal shape. Acetylcholinesterase catalytic site has been well established, consisting of a narrow gorge, about 20 Å long [1, 2] . The Peripheral Anionic Site (PAS), which corresponds to the entrance of the gorge, initially was believed to contain several negatively charged amino acids due to its preference of binding cationic ligands [3] . Subsequently, the enzyme crystal structure indicated that not sufficient acidic amino acids were located to this ligand-binding cavity to support the hypothesis [2-5]. Likewise, recent evidence suggests that the PAS is involved in the amyloid-β (Aβ) aggregation process observed in Alzheimer’s disease. The Catalytic Site (CAS), where the catalytic triad Ser200, Glu327 and His440 can be found, is in charge to inactivate acetylcholine by hydrolysis. Therefore, although acetylcholine could binds to the PAS, rapidly diffuses down to the catalytic site [6] . At the same time, some known inhibitors, such as tacrine and donepezil also interacts with the PAS [7] , but as acetylcholine they should diffuse to the CAS. In this sense, it is possible that selected compounds tested in our docking assays could interact with PAS, notwithstanding the importance of CAS and their amino acids (the catalytic triad and other residues, such as Glu199 and Trp84), should be more contributing to the AChE inhibition. Given the above, docking assays only into CAS should be an accurate mode to predict the inhibitory activity.

References

  1. Lizama, C., et al., Analysis of Carotenoids in Haloarchaea Species from Atacama Saline Lakes by High Resolution UHPLC-Q-Orbitrap-Mass Spectrometry: Antioxidant Potential and Biological Effect on Cell Viability. Antioxidants, 2021. 10(8): p. 1230.
  2. Sussman, J.L., et al., Atomic structure of acetylcholinesterase from Torpedo californica: a prototypic acetylcholine-binding protein. Science, 1991. 253(5022): p. 872-879.
  3. Nolte, H.-J., T.L. Rosenberry, and E. Neumann, Effective charge on acetylcholinesterase active sites determined from the ionic strength dependence of association rate constants with cationic ligands. Biochemistry, 1980. 19(16): p. 3705-3711.
  4. Raves, M.L., et al., Structure of acetylcholinesterase complexed with the nootropic alkaloid,(–)-huperzine A. Nature structural biology, 1997. 4(1): p. 57-63.
  5. Dougherty, D.A. and D.A. Stauffer, Acetylcholine binding by a synthetic receptor: implications for biological recognition. Science, 1990. 250(4987): p. 1558-1560. 6. Berg, L., et al., Targeting acetylcholinesterase: identification of chemical leads by high throughput screening, structure determination and molecular modeling. PloS one, 2011. 6(11): p. e26039.
  6. Agatonovic-Kustrin, S., C. Kettle, and D.W. Morton, A molecular approach in drug development for Alzheimer’s disease. Biomedicine & Pharmacotherapy, 2018. 106: p. 553- 565.

paragraph 3.4: no normal cell lines were considered to assess selective toxicity. No reference drug was used.

R:  We acknowledge the hint. However, we should consider that the test samples are extracts submitted to an initial bioprospecting protocol, which is aimed to identify molecules will potential pharmacological benefits. Thus, at this initial stage of the bioprospection the requested studies on extracts will provide meaningless results. Those studies will be necessary when the pure compounds are isolated and tested independently. This is routinely done as part of our screening protocol. J. Nat. Prod , https://pubs.acs.org/doi/10.1021/acs.jnatprod.5b00954 Reference compound values cisplatin was added in a table in supp material. We are also planning isolation of alkaloids from this plant and test selective toxicity of the pure compounds in a future work.

Reviewer 2 Report

The manuscript entitled «Alkaloid Profiling, Anti-Enzymatic and Antiproliferative Activity of
The Endemic Chilean Amaryllidaceae: Phycella cyrtanthoides»
is an orginal study realized on bulbs and leaves alkaloid extracts of this plant, analyzed using ultrahigh performance liquid chromatography orbitrap mass spectrometry (UHPLC-Orbitrap-MS) and tested for biological activities : enzyme inhibition potential against cholinesterases (acetylcholinesterase, butyrylcholinesterase and tyrosinase ), and antiproliferative effects against tumor cell lines (human cervical (HeLa), lung (A549, SW1573), colon (WiDr), and breast (HBL-100, T-47D). The subject was well-documented and experiment design was smartly conceived and perfectly performed and so yielding convincing conclusions. But minor revision is required to improve the manuscript. below are a few comments and concerns:

Some spelling errors troughout the manuscript should be corrected by carefully checking the whole text, below some examples of errors on words to put in italics :

- Pg 15, Ln 280 : Please put in italic « in vitro »

- Pg 15, Ln 280, Ln 284, Ln 289 : Please put in italic « Phycella. cyrtanthoides»

- Pg 15, Ln 286 : Please put in italic « P. herbetiana »

- Pg 15, Ln 287 : Please put in italic « Phycella »

So, I recommend the acceptation of this article « Metabolites » journal after these minor required revision.

Author Response

(The authors gave the same response as above.)

Reviewer 3 Report

Dear authors,

The manuscript entitled "Alkaloid Profiling, Anti-Enzymatic and Antiproliferative Activity of The Endemic Chilean Amaryllidaceae: Phycella cyrtanthoides” was to investigate the alkaloid profiling from the endemic Phycella cyrtanthoides from bulbs and leaves organs. It presents scientific relevance for the area of Plant Science and Medicine area.

After consulting www.sciencedirect.com and https://pubmed.ncbi.nlm.nih.gov/, publications were found for some authors involving the theme. However, you need to change some details/information in the Introduction, Material and Methods, Results, discussion and “conclusions”.

Introduction - It is well written, but I suggest:

-line 68: Remove double space.

-Some paragraphs are too short! I suggest joining them, for better understanding and continuity of ideas.

- As the article involves, also, the development and validation of the analytical method, I suggest including information on analytical techniques and more references on the determination of Alkaloid profiling and in biological samples (plants in general, or for the reported species) by UHPLC–DAD–MS.

-I suggest at the end of the introduction, I suggest highlighting the "innovative" proposal of the method, as well as the advantages/disadvantages.

Plant Material

- Page 3, “Protocol for obtaining extracts” section: Was this protocol based on any references? If yes, add!

-line 132, 140, 158, 155, 166, 168: °C

-line 174: Remove yellow mark

Results section:

-page 278: Add abbreviations L, B according to table 1.

- line 175, 280, 293, 360, 412: to write “Phycella cyrtanthoides” in italic and, throughout the manuscript!

-line 342: .

- Table 3: why the values are marked with "-".

Conclusion: Adequate, but I suggest to indicate disadvantages/limitations of the method and the study! Perhaps, to highlight the text in the 'Limits of the study' section.

Author Response

(The authors gave the same response as above.)

Reviewer 4 Report

line 57: italicize "Phycella"

text in general: galantamine or galanthamine?

text in general (also captions): P. cyrtanthoides and P. herbetiana must be italicized

line 135: change to "as follows"

lines 178 and 297 and 360: Torpedo californica

line 208: some alkaloids? only one (lycorine)

lines 241 and 270: change to "peaks"

line 262: change to "These alkaloids"

lines 315-318: THIS CONCEPT IS TOO SPECULATIVE without a proper quantification of the single components how is it possible to know if there aren't antagonistic effects or if the alkaloids compete for the same binding site?

docking studies: only CAS in AChE was considered? what about PAS?

paragraph 3.4: no normal cell lines were considered to assess selective toxicity. No reference drug was used.

Author Response

(The authors gave the same response as above.)
